# The Endocannabinoid System of the Nervous and Gastrointestinal Systems Changes after a Subnoxious Cisplatin Dose in Male Rats

**DOI:** 10.3390/ph17101256

**Published:** 2024-09-24

**Authors:** Yolanda López-Tofiño, Mary A. Hopkins, Ana Bagues, Laura Boullon, Raquel Abalo, Álvaro Llorente-Berzal

**Affiliations:** 1Department of Basic Health Sciences, University Rey Juan Carlos (URJC), 28922 Alcorcón, Spain; yolanda.lopez@urjc.es (Y.L.-T.); ana.bagues@urjc.es (A.B.); 2High Performance Research Group in Physiopathology and Pharmacology of the Digestive System (NeuGut-URJC), University Rey Juan Carlos (URJC), 28922 Alcorcón, Spain; 3Department of Pharmacology and Therapeutics, School of Medicine, University of Galway, H91W5P7 Galway, Ireland; lboullon@iu.edu (L.B.); m.hopkins9@universityofgalway.ie (M.A.H.); 4Galway Neuroscience Centre, University of Galway, H91W5P7 Galway, Ireland; 5Centre for Pain Research, University of Galway, H91W5P7 Galway, Ireland; 6High Performance Research Group in Experimental Pharmacology (PHARMAKOM-URJC), University Rey Juan Carlos (URJC), 28922 Alcorcón, Spain; 7Associated R+D+i Unit to the Institute of Medicinal Chemistry (IQM), Spanish National Research Council (CSIC), 28006 Madrid, Spain; 8Working Group of Basic Sciences on Pain and Analgesia, Spanish Pain Society, 28046 Madrid, Spain; 9Working Group of Basic Sciences on Cannabinoids, Spanish Pain Society, 28046 Madrid, Spain; 10Department of Physiology, School of Medicine, Autonomous University of Madrid (UAM), 28049 Madrid, Spain

**Keywords:** chemotherapy, endocannabinoid system, central nervous system, dorsal root ganglia, gastrointestinal, cisplatin

## Abstract

**Background/Objectives:** Cisplatin, a common chemotherapy agent, is well known to cause severe side effects in the gastrointestinal and nervous systems due to its toxic and pro-inflammatory effects. Although pharmacological manipulation of the endocannabinoid system (ECS) can alleviate these side effects, how chemotherapy affects the ECS components in these systems remains poorly understood. Our aim was to evaluate these changes. **Methods:** Male Wistar rats received cisplatin (5 mg/kg, i.p.) or saline on day 0 (D0). Immediately after, serial X-rays were taken for 24 h (D0). Body weight was recorded (D0, D1, D2 and D7) and behavioural tests were performed on D4. On D7, animals were euthanized, and gastrointestinal tissue, dorsal root ganglia (DRGs) and brain areas were collected. Expression of genes related to the ECS was assessed via Rt-PCR, while LC-MS/MS was used to analyse endocannabinoid and related N-acylethanolamine levels in tissue and plasma. **Results:** Animals treated with cisplatin showed a reduction in body weight. Cisplatin reduced gastric emptying during D0 and decreased MAGL gene expression in the antrum at D7. Despite cisplatin not causing mechanical or heat sensitivity, we observed ECS alterations in the prefrontal cortex (PFC) and DRGs similar to those seen in other chronic pain conditions, including an increased CB_1_ gene expression in L4/L5 DRGs and a decreased MAGL expression in PFC. **Conclusions**: A single dose of cisplatin (5 mg/kg, i.p.), subnoxious, but capable of inducing acute gastrointestinal effects, caused ECS changes in both gastrointestinal and nervous systems. Modulating the ECS could alleviate or potentially prevent chemotherapy-induced toxicity.

## 1. Introduction

There is a great number of chemotherapy drugs currently used in cancer treatment, including alkylating agents, vinca-alkaloids, antimetabolites, taxanes, etc. [1]. All these drugs induce cytotoxic activity in the tumour cells, but they can also affect healthy tissues, inducing side effects that decrease the quality of life of cancer patients, leading to treatment discontinuance [2,3,4].

Cisplatin is a chemotherapeutic agent currently used to treat different types of cancer, including sarcomas, melanomas, lymphomas, and different types of solid and soft tissue cancers [5,6,7]. Like other platinum-based alkylating agents, it forms intrastrand and interstrand crosslinks restricting DNA replication and transcription, which subsequently induces cell cycle arrest and programmed cell death [5,6,8]. Cisplatin is well known to induce several severe side effects related to its toxic and pro-inflammatory activity in the gastrointestinal and nervous systems that might reduce its clinical usefulness, on the one hand, while on the other hand, it can produce a dose-limiting effect and, therefore, a reduction in the effectiveness of the therapy [5,6]. These side effects can occur immediately after the first administration; for instance, nausea and vomiting start 1–4 h after treatment administration [5]. On the other hand, neurological alterations in cancer patients treated with cisplatin, like chronic neuropathic pain, tend to appear weeks after starting the treatment, although some may appear even after a single dose [9]. These early side effects are thought to be related to direct actions of cisplatin on the tissue and to an excessive release of pro-inflammatory mediators after the activation of the immunological system [3,6]. Another plausible reason for an early onset of cisplatin-related side effects is the accumulation of the drug in some organs, thus increasing the time that the antitumour agent is in contact with the healthy tissue. For instance, Breglio and cols (2017) [10] showed that, in mice, there was a significant peak concentration of platinum levels just one hour after the systemic administration of cisplatin in all the analysed tissues. Interestingly, in another analysis ten days after the end of the first treatment cycle, and just before the beginning of the second cycle, platinum levels were still detected in tissues such as the liver, kidney or cochlea [10].

Pharmacological activation of the endocannabinoid system (ECS) has shown to reduce chemotherapy-induced nausea/vomiting and neuropathy in preclinical studies (see [2] for a review). Moreover, two cannabinoid agonists, dronabinol and nabinol, are currently used for the treatment of nausea/vomiting in cancer patients and neuropathic pain [11,12]. Constituents of the ECS such as endocannabinoid ligands, i.e., anandamide (AEA) and 2-arachidonoylglycerol (2-AG); type 1 and 2 cannabinoid receptors (CB_1_ and CB_2_, respectively); and the enzymatic machinery necessary for endocannabinoid synthesis and degradation are expressed in the gastrointestinal and nervous systems [3,13,14,15,16]. In these tissues, endocannabinoid-related N-acylethanolamines such as palmitoylethanolamine (PEA) and oleoylethanolamine (OEA) are also expressed. They do not directly bind to CB_1_ or CB_2_ but bind to peroxisome proliferator-activated receptors (PPARs), among others [13,14,15,17,18,19]. PPARα activation has been shown to mediate anti-inflammatory effects in both the nervous [18] and gastrointestinal [19] systems. The main function of the ECS is related to the maintenance of the body’s homeostasis, and it is found to be altered under pathological conditions [13,14,15,17]. Despite the fact that it has been well established that pharmacological modulation of the ECS induces beneficial effects against chemotherapy-induced nausea/vomiting and neuropathic pain [2,3,13,14,15], little is known about how a chemotherapeutic drug can alter the expression of elements of the ECS in the gastrointestinal and nervous systems.

In view of this, we treated male rats with a single dose of cisplatin and analysed early development of chemotherapy-related gastrointestinal and neuropathic side effects. One week after cisplatin treatment, the levels of endocannabinoids (AEA and 2-AG) and related N-acylethanolamines (OEA and PEA), as well as mRNA expression of the endocannabinoid degrading the enzymes fatty acid amide hydrolase (FAAH) and monoacylglycerol lipase (MAGL), the two cannabinoid receptors (CB_1_ and CB_2_), and PPARα were measured in different structures of the gastrointestinal (including gastric antrum and fundus, ileum and distal colon) and nervous [prefrontal cortex (PFC), periaqueductal grey (PAG), amygdala (AMY) and L4 and L5 DRGs] systems. The levels of endocannabioids and related N-acylethanolamines were also determined in the plasma of these animals to analyse their potential usefulness as early biomarkers of chemotherapy-related toxicity.

## 2. Results

### 2.1. Body Weight and Intakes

Body weight, as well as food and water intakes, were recorded at different time-points along the study. Cisplatin-treated animals showed a reduction in body weight when compared with control animals and this difference reached statistical significance one week after cisplatin administration. Similarly, cisplatin significantly reduced food intake and this effect was statistically significant one week after its acute administration. However, cisplatin failed to induce any change in water intake during all the experimental timeline (Figure 1).

In the intra-subject effect analysis, the animal body weight’s repeated measures ANOVA revealed a significant effect of Time (F_1.341,13.408_ = 14.244) and Time–Treatment interaction (F_1.341,13.408_ = 20.499), but no significant overall effect of Treatment was observed in the inter-subject analysis. A further pair-wise analysis per time-point showed that cisplatin induced a significant reduction in body weight compared to saline-treated animals only at day 7 (D7): t_10_ = 2.840 (Figure 1A).

The repeated measures ANOVA of the grams of food consumed per animal in the home cage revealed a significant overall effect of Treatment (F_1,2_ = 606.619). A subsequent Kruskal–Wallis analysis revealed a lower food consumption of animals treated with cisplatin than controls only at one week after chemotherapy administration: D7: H_1_ = 3.971 (Figure 1B). No significant effects were observed in the water intake (mL) per animal (Figure 1C).

### 2.2. Semiquantitative Analysis of Gastrointestinal Motor Function

Gastrointestinal motility was determined for the first 24 h after drug administration using serial X-rays (see Section 4.4. for further information). Our results show that acute cisplatin treatment, with a single 5 mg/kg dose, induced a significant delay of gastric emptying during the first 24 h after drug administration that influenced the amount of barium reaching the caecum and the colon (Figure 2).

In brief, repeated measures ANOVAs revealed a significant intra-subject effect of Time in the stomach (F_2.487,24.875_ = 60.314), small intestine (F_2.869,28.685_ = 119.793), caecum (F_2.896,28.963_ = 136.963), and colon (F_2.961,29.609_ = 75.119) and of the Time–Treatment interaction in the stomach (F_2.487,24.875_ = 6.579), caecum (F_2.896,28.963_ = 8.553), and colon (F_2.961,29.609_ = 3.986). The inter-subject effect analysis of these repeated measures ANOVAs found a significant overall effect of Treatment in the stomach (F_1,10_ = 34.771) and colon (F_1,10_ = 6.713). A further pair-wise analysis in each time-point of the stomach showed that cisplatin significantly reduced gastric motility 2 h after drug administration, until the end of the analysis (time-point 24 h): Time 2 h t_10_ = −5.590; Time 3 h t_10_ = 5.721; Time 4 h t_5.973_ = 3.893; Time 6 h t_10_ = 2.289; Time 8 h H_1_ = 7.333; Time 24 h H_1_ = 7.500. In the colon, we observed a significantly lower amount of barium reaching the colon from 4 to 8 h after cisplatin treatment: Time 4 h H_1_ = 4.275; Time 6 h t_10_ = 2.907; Time 8 h t_6.389_ = 3.630. However, 24 h after drug administration, levels of barium tended to be higher in cisplatin-treated than saline-treated animals (Figure 2): Time 24 h t_10_ = 1.941; *p* = 0.081.

### 2.3. Behaviour

Different behaviour tests were performed to analyse general locomotor activity as well as nociceptive thresholds. The behavioural data were analysed using an unpaired *t*-test. The results revealed no significant effects on locomotor activity (number of beam interruptions; Figure 3A), mechanical tactile sensitivity (von Frey test, Figure 3B), or heat tactile sensitivity (Hargreaves’ test, Figure 3C).

### 2.4. Weight and Macroscopic Analysis of Organs

Seven days after cisplatin treatment, a macroscopic study (weight and/or dimension analysis) of several gastrointestinal regions and the kidneys was performed. Data analysis in the gastrointestinal tissue revealed a statistically significant decrease in the weight of the stomach (t_5.24_ = 3.35) and the small intestine [full (t_10_ = 4.15), empty (t_10_ = 4.12) and its contents after its milking (t_10_ = 2.97)] of cisplatin-treated rats compared with the saline-treated group (Table 1).

In addition, cisplatin also decreased the size of the stomach (t_6.09_ = 3.22) but did not change that of the other gastrointestinal regions or the weight of the kidneys (Table 1).

### 2.5. Levels of Endocannabinoid Ligands and Related N-Acylethanolamines

One week after cisplatin administration, plasma and several gastrointestinal and nervous systems regions were collected and the levels of endocannabinoid ligands (AEA and 2-AG) and related N-acylethanolamines (PEA and OEA) were measured using liquid chromatography coupled to tandem mass spectrometry (LC-MS/MS).

Data analysis in the gastrointestinal tissue only revealed a significant increase in the levels of AEA in the ileum of cisplatin-treated rats compared with their saline-treated counterparts: t_8.403_ = −2.308 (Table 2).

No significant differences were observed when levels of endocannabinoid and related N-acylethanolamines were analysed in CNS tissue and plasma of the experimental animals (Table 2).

### 2.6. Gene Expression Results

Reat-time PCR was used to analyse the expression of genes related to the ECS, such as *cnr1* (CB_1_), *cnr2* (CB_2_), *faah* (FAAH), *mgll* (MAGL), and *ppara* (PPARα), in the gastrointestinal and nervous tissues collected from saline- and cisplatin-treated male rats.

No significant differences were observed when gastrointestinal tissues of cisplatin and saline-treated rats were analysed for CB_1_, CB_2_, and PPARα receptors. Regarding the gene expression of catabolic enzymes, no significant differences were observed for FAAH, but we observed a significant decrease in the expression of MAGL in the antrum (t_9_ = 2.528) in cisplatin-treated animals compared with controls (Figure 4), without significant differences in the other gastrointestinal regions analysed (Appendix A).

Gene expression analyses in L4 and L5 DRGs revealed a significant increase in the gene expression of CB_1_ in L4 (t_7_ = −2.451; Figure 5A) and a trend in L5 (t_10_ = −1.833; *p* = 0.097; Figure 5B) after cisplatin treatment, with no further changes in the other genes analysed (Appendix A). The analysis of gene expression differences induced by cisplatin treatment in the central nervous system revealed a significant increase in the gene expression of CB_2_ (t_9_ = −4.747; Figure 5C), MAGL (t_9_ = −2.680; Figure 5D), and PPARα (t_9_ = −3.567; Figure 5E) in the PFC. The analysis of the PAG did not reveal any significant difference (Appendix A), but we observed a significant decrease in CB_1_ (t_9_ = 2.349; Figure 5F) and MAGL (t_8_ = 3.077; Figure 5G) gene expression in the AMY of cisplatin-treated animals compared with their saline counterparts.

## 3. Discussion

In this study, we analysed the evolution of body weight and food/water intake, acute gastrointestinal motor function, early development of pain-related behaviours, and lastly the expression of components of the ECS in gastrointestinal and nervous tissues after treatment with a 5 mg/kg dose in adult male rats. In the clinic, cancer patients are usually treated with repeated administrations of chemotherapeutic drugs, and therapeutic approaches directed to reduce or avoid the appearance of chemotherapy-related side effects should be preventively administered some time before, or shortly after the first (and each) administration of chemotherapy. Our goal was to investigate short-term effects induced by a chemotherapy agent shortly after its first administration and immediately before the second one would be given. To do this, we treated our animals with a dose of cisplatin aimed to induce neuropathic pain [20] and analysed the alterations induced by an acute administration of the drug for a week. Moreover, levels of endocannabinoid ligands and related N-acylethanolamines were measured in plasma to investigate the potential usefulness of these parameters as early biomarkers of chemotherapy-induced toxicity.

Cisplatin is known to exert many different toxic effects [21], some of them caused by many antitumour drugs and others quite specific to this drug. In this study, cisplatin significantly reduced body weight one week after treatment, an effect previously shown after acute and repeated treatment with diverse doses of cisplatin [22,23] and caused by many antitumour drugs [24]. One specific effect induced by cisplatin is renal damage [25,26], but we did not appreciate this toxicity simply from the weight and macroscopic appearance of this organ, nor upon microscopic evaluation, probably because of the short, acute treatment used.

Different mechanisms could underlie the reduction in body weight. First, cisplatin reduced food intake, which may account for the reduced size and weight of the stomach and small intestine and its reduced contents at sacrifice (Figure 2B and Table 2). Second, cisplatin-related mucositis and/or diarrhoea could also decrease body weight by decreasing the absorption of nutrients. Indeed, we have previously reported damage to the gastrointestinal mucosa after a single high dose [27] and repeated treatment with cisplatin [28]. However, diarrhoea did not seem to occur in our animals, since no signs were observed either clinically or on the radiographs obtained at different time-points after administration (Figure 2), which is in accordance with a previous study where a higher single dose (6 mg/kg) of cisplatin was administered to the rats [27]. Furthermore, there was a lack of significant alterations in water intake (Figure 2C), which could have been increased if excessive/sudden loss of water occurred through the faeces. A reduction in body weight could also be due to increased locomotor activity, but this was not observed here four days after cisplatin administration, whereas four weekly administrations of 5 mg/kg induced a significant reduction in horizontal and vertical activity in male Sprague Dawley rats [29]. Thus, the most likely explanation for the reduced body weight is reduced food intake.

Our results show a significant reduction in gastric emptying, in concordance with previous studies from our group, following both acute and repeated administration of cisplatin [22,23,27]. These effects may contribute to cisplatin-induced anorexia [30]. Furthermore, cisplatin administration is usually associated with the development of nausea and vomiting in humans [5,31] and with indirect markers of nausea in animals that cannot vomit, such as pica or gastric distension in rats [2,22,23,32]. Indeed, gastric distension was observed both in this study for a single dose of 5 mg/kg (Figure 2E) and previously when a 6 mg/kg of cisplatin was administered [23,27]. Cisplatin is known to induce a massive release of serotonin from enterochromaffin cells, which activates 5-HT3 receptors on vagal afferent fibres and enteric nerves inducing a delay of gastric emptying and an increase in food retention in the fundus of the stomach [27,33,34]. This explains the efficacy of 5-HT3 antagonists as a first-line treatment of acute chemotherapy-induced nausea and vomiting, particularly when cisplatin is used [35]. However, other neurotransmitters may also participate in the development of gastric dysmotility, nausea, and vomiting, particularly at later times [35]. Interestingly, one week after administration, despite no changes in the levels of 2-AG being observed in the antrum, gene levels of its catabolizing enzyme, i.e., MAGL, were significantly lower in cisplatin-treated rats than their saline counterparts. The discordance between the levels of 2-AG and the gene expression of its catabolic enzyme may be due to the fact that the molecular analysis of gastrointestinal tissue was carried out in whole gut wall samples, while the increase in 2-AG induced by a lower expression of MAGL might be present only in the myenteric plexus, the area intrinsically involved in gastric emptying. On the other hand, it could be that alterations in 2-AG levels have occurred due to rapid metabolic modifications occurring postmortem, as suggested by other authors in brain structures [36,37]. Whatever the case may be, this lower expression of MAGL may favour an increase in CB_1_ receptor signalling in the antrum and a subsequent decrease in gastric emptying [38,39].

Cisplatin did not alter the transit of the small intestine and caecum during the X-ray session. This agrees with our previous studies showing that acute treatment with 2 mg/kg of cisplatin failed to alter small intestine motility [28]. However, after a weekly administration of this dose of cisplatin (cumulative dose of 10 mg/kg), we observed a slower small intestinal transit, accompanied by important histological and cellular alterations [28], as well as the development of an enteric neuropathy, also observed in the large intestine [28,40]. Interestingly, we also observed a significant increase in the levels of AEA in the ileum of animals treated with cisplatin. We have previously reported that different cannabinoid agonists reduce motor function in the small intestine and that this effect is reversed by CB_1_ (but not CB_2_) antagonists [38]. Likewise, administration of an FAAH inhibitor also reduced intestinal motility in a CB_1_-dependent manner [41]. Therefore, the increase in AEA in the small intestine one week after treatment supports the theory that cisplatin may actually reduce small intestinal motility—if not immediately after its administration, later in time [27,42]—and that this effect might be mediated by increased activity of the intestinal CB_1_ receptors through increased AEA signalling.

The radiographic study showed a significant reduction in the colonic contents in cisplatin-treated rats. Although the effect observed in the colon could very likely be due to the delayed gastric emptying, which would decrease the arrival of barium-stained gut contents to the colon, a decrease in the colonic motility per se cannot be discarded. In these regards, one week after repeated cisplatin treatment (cumulative dose of 10 mg/kg), no colonic dysmotility was observed in vivo [28], but colonic contractility was altered in vitro [43], likely due to the mentioned development of an enteric neuropathy [40]. If colonic contractility is also altered one week after a single 5 mg/kg dose of cisplatin is not known, but—if present—such an effect would not likely involve the ECS, since no significant changes in the expression of its components were found here.

The behavioural analysis showed that cisplatin failed to induce statistically significant mechanical and heat hypersensitivity 4 days after treatment. We have previously reported that repeated weekly administration of 2 mg/kg of cisplatin induces significant mechanical hypersensitivity in rats after 4 weeks [22] and after 5 weeks [44]. To the best of our knowledge, only one study analysed the development of pain-related hypersensitivity of a single dose of 5 mg/kg of cisplatin in the rat. In this study, Nayebi and cols. (2012) observed thermal hypersensitivity on day 5 after treatment, although the highest thermal hypersensitivity was observed on day 15 [20]. In contrast, our results in the Hargreaves’ test show a lack of thermal hypersensitivity in our animals 4 days after cisplatin administration. These differences may be due to several factors, such as the different post-treatment time of testing, the different anatomic structures of the animal where the heat stimulus was applied (tail vs. hind paw) or to the different intensity of the heat source. In the present study, the dose used was unable to induce not only thermal but also mechanical hypersensitivity, without changes in locomotor activity, so under our experimental design, the development of chemotherapy-induced neuropathic pain may require a higher dose of cisplatin and/or more time to sensitise the pain pathways.

In this study, we evaluated changes in the ECS of L4 and L5 DRGs. Due to the small size of the tissue, we prioritised the analysis of mRNA expression over the biochemical analysis, so we could not analyse the levels of endocannabinoid ligands and related N-acylethanolamines in this tissue. Khasabova and cols. (2014) [45] did analyse the levels of these molecules after daily administration of cisplatin (1 mg/kg) in mice for a week, and they observed a significant decrease in AEA and 2-AG. These changes were mainly due to alterations in the gene expression levels of their catabolic enzymes (FAAH and MAGL, respectively). However, instead of increasing them, cisplatin significantly reduced mRNA expression of FAAH and MAGL. The authors suggested that a feedback response and/or a direct response to the effect of cisplatin may be responsible for these contradictory results [45]. Guindon and cols. (2013) analysed the effect of a weekly dose (for 3 weeks) of 3 mg/kg of cisplatin on the mRNA expression of cannabinoid receptors and catabolic enzymes in rat DRGs [46]. While we observed a significant increase in the mRNA expression levels of CB_1_ in L4 and a trend in the same direction in L5, these authors did not find any significant difference [46]. Curiously, in another report using the spinal nerve ligation model of neuropathic pain (at the level of L5) in rats, a significant increase in the expression levels (both at mRNA and protein levels) of CB_1_ was observed in the ipsilateral L4 without changes in L5 [47]. This is interesting since the authors were expecting an increase in this cannabinoid receptor in L5, where the lesion was made, and not L4. On the other hand, they observed a significant increase in the levels of AEA and 2-AG in L5, but not in L4 [47]. The authors interpreted these conflicting results as an adaptation to the hyperexcitability induced by the surgery in the ipsilateral side. This hyperexcitability would induce an increase in CB_1_ in the DRGs, in both L4 and L5. However, in the ipsilateral L5, and due to the neuroinflammation observed in this area, there is also an increase in AEA and 2-AG that would reduce the expression of CB_1_ in L5, but not in L4, where the levels of AEA and 2-AG were the same as in controls [47]. In view of this, we could suggest that our cisplatin treatment induced a hyperexcitability in sensory neurons of the DRGs, which induced a significant increase in the expression of CB_1_. However, as our treatment was less harmful than a direct ligation to the nerve, there was no neuroinflammation and no increase in the levels of AEA and 2-AG of the DRGs, and subsequently no decrease in the already overexpressed CB_1_. Obviously, biochemical analysis of these samples would have given us a better perspective of this theory.

A single dose of cisplatin 5 mg/kg failed to induce any change in the levels of endocannabinoid ligands and related N-acylethanolamines in the PFC, PAG, or AMY one week after its administration. In line with this result, an acute treatment with 1 mg/kg did not alter levels of AEA, 2-AG, and PEA in the midbrain of male mice one week after treatment [45]. Remarkably, only one study has analysed the levels of endocannabinoid ligands and related N-acylethanolamines in the nervous system after repeated administration of cisplatin in rats. In this study, from Guindon and cols. (2013) [46], the authors observed that a treatment with a dose of 3 mg/kg/week for 3 weeks induced a significant increase in the levels of AEA and 2-AG in the lumbar spinal cord. The authors suggested that this increase happens as an adaptive central response to the peripheral neuropathy induced by cisplatin [46]. In view of this, we could hypothesise that our single dose, that was inefficient to induce the classic neuropathic pain-related mechanical hypersensitivity, did not elicit a response in the central nervous system (CNS), and hence did not alter the levels of endocannabinoid ligands and related N-acylethanolamines in brain regions involved in the descending pain pathway.

However, despite this lack of effects in the levels of endocannabinoids, we did observe interesting alterations in the gene levels of several elements of the ECS that may suggest that changes in the levels of endocannabinoids do happen, but only shortly after cisplatin administration, tending to return to baseline levels quickly after the initial insult in order to restore homeostasis into the CNS. The molecular analysis showed, for instance, that cisplatin induced a significant increase in the gene expression for MAGL in the PFC and a significant decrease in the AMY. Alterations in the corticoamygdalar pathway have been shown to play a pivotal role not only in pain but also in emotion–pain interactions [48,49,50]. In normal conditions, pyramidal cells (PCs) from the basolateral amygdala (BLA) send excitatory glutamatergic projections to PCs of the PFC that project to the central nucleus of the amygdala (CeA) inducing an inhibition of the outputs in this region. In brief, an activation of the PFC leads to a deactivation of CeA outputs. However, in acute animal models of arthritis, colitis, formalin-induced inflammation, and neuropathic pain, CeA output firing is increased (see [49] for more information) and mediates pain-related emotional responses [50]. The ECS modulates the corticoamygdalar pathway through the CB_1_ receptor in inhibitory GABAergic interneurons of the PFC. BLA activation of PCs in the PFC activates the production and release of 2-AG, which will bind to CB_1_ in GABAergic interneurons modulating GABA-related inhibition of the PCs [48,50]. Interestingly, in arthritic animals there is a lack of this ECS-mediated control, and the mechanism involved seems to be related to an increase in 2-AG hydrolysis rather than changes in the activity/expression of CB_1_ [51]. Our results show that cisplatin induced a significant increase in MAGL in the PFC, without changes in the expression levels of CB_1_. This increase in MAGL would induce a decrease of 2-AG that cannot modulate the feedforward inhibition induced by the GABAergic interneurons, decreasing PFC-related firing to the CeA. Since PFC-related control of CeA is through inhibitory interneurons, the decrease in PFC firing would induce an increase in CeA outputs and an increase in pain-related emotional responses (Figure 6). Still, we did not detect any changes in the levels of 2-AG in our biochemical analysis, and we did not evaluate emotionality in these animals, so further analyses are needed to confirm this hypothesis.

Regarding the decrease in MAGL mRNA expression in the AMY, the metabolic activity in PFC and AMY is often inversely correlated in clinical populations with an imbalance in the corticoamygdalar pathway [48]. So, it is plausible that the decreased activity of the ECS in the PFC (shown by an increase in the hydrolytic enzyme MAGL) may be reflected as increased activity (lower levels of MAGL) in the AMY. Despite the fact that no changes in the levels of 2-AG were observed in the AMY, it is possible that the hyperactivation of the ECS is behind the decreased levels of mRNA for CB_1_ also observed in the AMY of cisplatin-treated animals, as a way to counteract an excessive activation of CB_1_. Anyway, and due to the differential activity of the two nuclei of the AMY (BLA vs. CeA) and the fact that we analysed mRNA expression in the whole AMY, we should be very careful when interpreting these results.

In the PFC, the increased gene expression of MAGL was accompanied by an increased gene expression of CB_2_ and PPARα. It is well established that CB_2_ expression increases due to neuroinflammatory processes [52] and that a decrease in MAGL and an increase in PPARα signalling have anti-inflammatory and neuroprotective activities in the CNS [53]. Neuroinflammation in the PFC usually leads, when persistent, to cognitive impairments, and due to the pro-inflammatory nature of chemotherapy action, it is well known that chemotherapeutic drugs can induce cognitive deficits in a process known as chemotherapy-induced cognitive impairment or chemobrain [2,54,55]. In fact, it has been observed that a repeated weekly treatment with a 5 mg/kg dose of cisplatin induces cognitive deficits, hippocampal activation of NF-κB and expression of downstream inflammatory mediators in rats [56], and a decreased percentage of alternations in the Y maze, indicative of a poorer hippocampal spatial memory, accompanied by pro-inflammatory and proapoptotic hippocampal markers [29]. In view of this, we could argue that the alterations in the gene expression of CB_2_ and PPARα in the PFC of cisplatin-treated rats may be related to acute neuroinflammatory processes that may have induced cognitive deficits in these animals. A future study should analyse the cognitive effects of an acute administration of 5 mg/kg of cisplatin.

Finally, we also measured the levels of endocannabinoid ligands and related N-acylethanolamines in plasma to determine the potential usefulness of these molecules as an early biomarker of chemotherapy-related neuropathic pain or gastrointestinal function. Several studies have shown that plasma/serum levels of these lipidic compounds can be used as biomarkers of chronic pain syndromes such as osteoarthritis pain [57], complex regional pain syndrome [58], fibromyalgia [59,60], migraine [61], and interstitial cystitis/bladder pain syndrome [62]. However, we did not observe any significant alteration in the circulating levels of AEA, 2-AG, PEA, and OEA in our plasma samples. Thus, circulating levels of these compounds seem to not be useful as an early biomarker of neuropathic pain induced by a chemotherapeutic treatment. In any case, at this stage we cannot discard the utility of these biomarkers in models of neuropathic pain induced by other chemotherapeutic drugs, such as oxaliplatin, vincristine, etc., or with cisplatin at different time-points or with a different dose regime. Regarding the possible usefulness of endocannabinoid ligands as markers of chemotherapy-induced gastrointestinal dysfunction, plasma levels of endocannabinoid ligands, such as AEA, 2AG, and PEA, were altered in women suffering from gastroparesis, a chronic gastrointestinal motility disorder where there is delayed gastric emptying without gastric outlet obstruction [63]. The authors also analysed the plasma levels of these endocannabinoids in men suffering from this condition, but in this case, no significant effect was observed, which is in line with our current results. A future study in which female rats are also evaluated is required to check if cisplatin-induced delayed gastric emptying may be able to induce, at least in females, alterations in plasma endocannabinoid levels and, hence, be useful as a biomarker of cisplatin-related gastrointestinal effects.

In summary, our results show that a single acute treatment with a 5 mg/kg dose of cisplatin induces important alterations in the ECS of gastrointestinal and nervous systems that could be linked to changes in gastrointestinal motility and early development of neuropathic pain, severe side effects observed in cisplatin-treated cancer patients.

## 4. Materials and Methods

### 4.1. Animals

Twelve male Wistar rats (366–430 g, 4-months-old) were obtained from the Veterinary Unit of URJC (Madrid, Spain), where the in vivo studies were performed. Animals were group housed (3–4/cage) in standard transparent cages (60 × 40 × 20 cm) with water and food (LASQ diet^®^ Rod 14-A www.altromin.de) available ad libitum. The animal holding room was maintained at a constant temperature (22 ± 0.5 °C) and humidity (55 ± 3%). All in vivo procedures were performed during the light phase (8:00–20:00) of the standard light conditions 12:12 h. From these animals, samples were obtained to perform ex vivo analyses at the University of Galway.

The experiments were designed and performed according to the EU Directive for the Protection of Animals Used for Scientific Purpose (2010/63/EU) and Spanish regulations (Law 32/2007, RD 53/2013 and order ECC/566/2015) and approved by the Ethical Committee at Universidad Rey Juan Carlos (URJC) and Comunidad Autónoma de Madrid (PROEX 142.8/21). The number of animals used and their suffering were minimised.

### 4.2. Drug Preparation and Dose Selection

Cisplatin was purchased from Merck Life Science (Darmstadt, Germany) and dissolved in saline before administration (sonicated for about 15 min). Saline or cisplatin volumes were adjusted to a maximum of 2.5 mL/kg.

The dose of cisplatin was chosen based on previous experience in our group. We have previously reported that acute administration of 6 mg/kg can induce gastrointestinal dysmotility [23,27]. However, this dose exhibits high lethality 3 days after administration, so we chose a slightly lower dose (5 mg/kg). On the other hand, we also aimed to analyse if early development of neuropathic pain-related behaviours and/or early alterations in the endocannabinoid system of the descending pain pathway appeared. In this sense, administration of 5 mg/kg of cisplatin was shown to induce heat hypersensitivity in the tail-flick test 15 days after treatment in rats [20].

### 4.3. Experiment Outline

Animals were randomly assigned to pharmacological treatment (n = 6), housed per experimental group, and habituated to the holding room conditions for a period of 4 days. On the first day of the study (day 0), the animals were injected either with saline (0.9% NaCl in distilled water) or cisplatin (5 mg/kg; i.p.). As illustrated in Figure 7, gastrointestinal motility was determined for the first 24 h after drug administration using serial X-rays. Food/water intake and body weight were recorded on days 0, 1, 2, and 7 after cisplatin administration. Intakes were recorded by home cage and divided by the number of animals each cage held. Locomotor activity (actimeter) and pain-related behaviours (von Frey and Hargreaves’ tests) were measured on day 4. On day 7, animals were sacrificed and several areas of the gastrointestinal tract (fundus, antrum, ileum, colon), L4 and L5 DRGs, and brain areas related to the descending pain pathway (PFC, PAG, and AMY) were collected, snap-frozen on dry ice, and kept at −80 °C until use. Plasma samples were obtained from trunk blood samples, aliquoted, and frozen at −80 °C for further analyses.

### 4.4. Radiographic Analysis of Gastrointestinal Motility

Gastrointestinal motor function was studied using a radiographic technique without prior fasting as previously described [23,27]. In brief, immediately after cisplatin or saline administration, animals were intragastrically administered with a radiopaque contrast consisting of a barium sulphate suspension (Barigraf, Juste SAQF, 2 g/mL, T = 22 °C). Plain facial radiographs (20 ms) were obtained using a CS2100 (Carestream Dental, Madrid, Spain) digital X-ray apparatus (60 kV, 7 mA) with a focus distance manually fixed to 50 ± 1 cm. X-rays were recorded on Carestream Dental T-MAT G/RA films (15 × 30 cm) housed in a hand-made cassette with a regular intensifying screen, immediately and 1, 2, 4, 6, 8, and 24 h (T0–T24) after contrast administration. The film cassette was placed beneath a restraining tube in which the animal was inserted. Films were developed in a Kodak X-omat 2000 automatic processor (Kodak AG, Stuttgart, Germany).

Gut motility was determined semiquantitatively using the images by assigning a compounded value to each region of the gastrointestinal tract considering the following parameters: percentage of the region filled with contrast (0–4); intensity of the contrast (0–4), homogeneity of the contrast (0–2), and sharpness of the profile of the gastrointestinal region filled with contrast (0–2). Each of these parameters was scored and a sum (0–12 points) was made per animal, time-point, and gastrointestinal region, which included the stomach, small intestine, caecum, and colon.

### 4.5. Macroscopic Analysis and Weight of Organs of Interest

After sacrifice, the size of the different gastrointestinal organs was evaluated. Gastrointestinal organs were removed en bloc, the intestines were straightened out on graph paper, and pictures of the gastrointestinal tract were taken. To analyse the size of the stomach and caecum and the length of the small intestine and colon, the pictures were analysed using Image J 1.38 (Windows, National Institute of Health, USA, free software: https://imagej.net/) and the graph paper was used to set the scale for each picture. The size of the stomach and caecum was estimated as the areas occupied by their 2D projections on the graph paper.

After the photo was taken, the stomach, small intestine, caecum, and colorectum were separated and weighed. Additionally, the small intestine and colon were emptied and then weighed again, along with the content of the small intestine. Since cisplatin causes nephrotoxicity, the weight of the kidneys was also recorded.

### 4.6. Behavioural Tests

On day 4 post-administration, animals were exposed to a range of behavioural tests to measure pain-related behaviours and locomotor activity.

#### 4.6.1. Von Frey Test

Mechanical hypersensitivity to non-noxious stimuli was assessed using a series of calibrated von Frey filaments, starting with 6 g and with a cut-off value of 26 g (EVF3, Bioseb^®^, Vitrolles, France), as previously described [44]. Animals were placed individually on an elevated wire mesh floor, covered by a transparent plastic cage (27 × 21 × 15 cm), and were allowed to adapt to the testing environment for 20 min, before testing. The filaments were inserted through the wired mesh floor and applied to the plantar surface of the hind paw. Each filament was applied five times on each paw. The mechanical threshold was recorded, corresponding to the minimum force necessary for the animal to withdraw its paws in three of the five trials. Measurements on the same paw were spaced apart, by at least 30 s. The results are expressed as the mean of the values obtained from both hind paws.

#### 4.6.2. Hargreaves’ Test

Hargreaves’ test was conducted to determine the sensitivity threshold to heat noxious stimuli as previously described [44]. Responses to thermal stimuli were evaluated immediately after the von Frey test using a Hargreaves 37,370 apparatus (Ugo Basile, Gemonio, Italy). Rats were placed separately on an elevated glass floor in a transparent cage (15 × 15 × 10 cm) and were allowed to adapt to the environment for 20 min, before testing started. The withdrawal latency from a focused beam of radiant heat applied to the mid plantar surface of the hind paws was recorded. The intensity of the light was adjusted at the beginning of the experiment so that the control average baseline latencies were about 8 s and a cut-off latency of 25 s was imposed. The withdrawal latency of each paw was measured during three trials separated by two-minute intervals, and the mean of three readings was used for data analysis.

#### 4.6.3. Locomotor Activity

Locomotor activity was evaluated using individual photocell activity chambers after somatic sensitivity assessment, as described previously [44]. Each rat was placed separately in the recording chamber (Cibertec S.A, Madrid, Spain; 55 × 40 cm; 3 cm spacing between beams) where the number of interruptions of photocell beams was recorded over a 30 min period. The mean number of crossings of the photocell beams was used for comparison.

### 4.7. Liquid Chromatography Coupled to Tandem Mass Spectrometry (LC-MS/MS)

#### 4.7.1. Tissue Lipid Extraction

Quantification of endocannabinoids (AEA and 2-AG) and related N-acylethanolamines (PEA and OEA) levels in gastrointestinal and nervous tissue was carried out following a lipid extraction method described previously [64,65] with some modifications. In brain tissue (PFC, PAG, and AMY), 500 μL of 100% acetonitrile (ACN) was added to each sample and the tissue was homogenised for ~6 s on ice using an ultrasonic homogeniser (Mason, Dublin, Ireland) followed by a 13,000 rpm centrifugation for 15 min at 4 °C in a Hettich centrifuge Mikro 22R (Hettich, Germany). On the other hand, 500 μL of 100% ACN was added to the gastrointestinal samples, except for the gastric antrum, in which 2 mL of 100% ACN was added. The homogenisation of the gastrointestinal tissue was achieved in two steps: (1) the tissue was homogenised using a mechanical homogeniser (IKA T10 Ultra-Turrax, Germany) for ~15 s, and (2) the resulting homogenate was homogenised again using an ultrasonic homogeniser for ~3 s with 3–4 repetitions separated at least for 5 min. The homogenisation of all the tissues was always conducted on ice and the tissues were kept at a constant temperature of 4 °C. Finally, the gastrointestinal tissue was centrifuged in the same conditions as the nervous tissues, except for the antrum, which was performed in a Hettich ROTINA 38/38R (Hettich, Germany). The supernatant from this centrifugation was kept on ice while the pellet was stored at −80 °C for further use. Two hundred microlitres of 100% ACN containing deuterated internal standards (DISs) for endogenous cannabinoid ligands (DIS mix: 2.5 ng d8-AEA, 50 ng d8–2-AG, 2.5 ng d4-PEA, 2.5 ng d4-OEA; Cayman Chemicals, Biosciences, Cambridge, UK) were added to seventy-five microlitres of supernatant. Finally, 40 μL of each sample was added to HPLC vials.

#### 4.7.2. Plasma Lipid Extraction

Frozen plasma samples were placed on ice for 30 min. Once thawed, plasma samples were centrifuged under the same conditions as nervous samples (Section 4.7.1.). Twenty microlitres of the DIS mix was added to two hundred microlitres of plasma and the mixture was then vortexed and allowed to equilibrate for 10 min on ice. Protein precipitation was obtained by adding 1 mL of 100% ACN containing 0.1% formic acid, maintained at 4 °C, to each sample, followed by incubation on ice for 30 min. The precipitated proteins were pelleted by centrifugation at 13,000 rpm for 15 min at 4 °C. A filter (FisherbrandTM Non-sterile PTFE Hydrophyl, 25 mm, 0.45 μM Syringe Filter, Fisher Scientific, Dublin, Ireland) was placed in a 5 mL SafeSeal tube (SARSTEDT, Wateford, Ireland) for each sample. Then, 1 mL syringes were attached to the filters by carefully inserting the syringe neck into the filter inlet. The plunger/piston from each 1 mL syringe was removed and saved in a sterile container or microfuge tube. A total of 1100 μL of the supernatant was removed and loaded into the syringe. The supernatant was allowed to flow through the filter until the syringe was emptied. Once the supernatant had flown into the microfuge tube, 700 μL of 100% ACN was added to the syringe to displace the dead volume of the filter. The syringe plunger was slowly re-inserted, and the 100% ACN was pushed through the filter. The collected eluate was vortex-mixed, and 500 μL of the total volume was dried down at 45 °C for approximately 1 h in a centrifugal concentrator (Eppendorf Concentrator plus complete system, Davidson and Hardy Ltd., Belfast, Ireland). Each sample was reconstituted in 40 μL of 100% ACN before transferring it to HPLC vials.

#### 4.7.3. Standard Curve

A 10-point and ¼ dilution standard curve was prepared, in which the highest standard contained 25 ng of AEA, PEA, and OEA and 250 ng of 2-AG per 25 μL of non-deuterated internal standards in a final volume of 75 μL. Finally, 200 μL of 100% ACN containing DIS mix was added to each standard curve point. Forty microlitres of each standard curve point was added to the HPLC vials.

#### 4.7.4. LC-MS/MS

The mobile phases consisted of (1) high-performance liquid chromatography grade water with 0.1% (*v*/*v*) formic acid and (2) ACN with 0.1% (*v*/*v*) formic acid for the initial three minutes with a flow rate of 0.2 mL/min using a Zorbax^®^ SB C18 column (1.8 μm particle dimension, 50 mm length, 2.1 mm internal diameter; Agilent, Santa Clara, CA, USA). Reversed-phase gradient elution was initiated at 45% solution B for 1 min, then ramped linearly up to 100% solution B for 4 min and held at 100% solution B until 12 min. At 12.1 min, when the assay run finished, the gradient returned to initial conditions for a further 4–5 min to re-equilibrate the column before the next injection. Under these conditions, AEA, 2-AG, PEA, and OEA eluted at the following retention times: 13.8 min, 14.1 min, 14.3 min, and 14.6 min, respectively. Analyte detection was carried out in electrospray-positive ionisation mode on an Agilent 1260 infinity 2 HPLC system SCIEX QTRAP 4500 mass spectrometer operated in triple quadrupole mode (SCIEX Ltd., Phoenix House Lakeside Drive Centre Park, Warrington, UK). Instrument conditions were optimised for each analyte by infusing standards separately. Quantitation of target endocannabinoids and related N-acylethanolamines was achieved by positive ion electrospray ionisation and multiple reaction monitoring (MRM) mode, allowing simultaneous detection of the protonated precursor and product molecular ions [M + H^+^] of the analytes of interest and the deuterated form of the internal standard. Quantitation of each analyte was performed by determining the peak area response of each target analyte against its corresponding deuterated internal standard. This ratiometric analysis was calculated using Skyline software v.21.2 (MacCoss Lab Software, Seattle, DC, USA). The amount of analyte in unknown samples was calculated from the interpolation of the relative response (i.e., the ratio of the peak area of the non-deuterated analyte to the peak area of the internal standard or deuterated analyte) in a 10-point calibration curve constructed from a relative concentration calculated from the ratio of the concentration of the non-deuterated analyte to a fixed amount of deuterated analyte or internal standard. Levels of endocannabinoids and related N-acylethanolamines were expressed in molar units per g of tissue or per μL of plasma.

### 4.8. Real-Time PCR

L4 and L5 DRGs’ total RNA was extracted directly from the dissected tissue, while in the other tissues, RNA was extracted from the pellet obtained after tissue homogenisation in Section 4.7.1. In both cases, RNA extraction was obtained using a Machery-Nagel extraction kit (Nucleo spin RNA II, Technopath, Dublin, Ireland) according to manufacturer’s instructions and as previously described [18,65,66]. RNA quality (1.8–2.0 as determined by λ260/λ280 ratio) and quantity (ng/μL) were assessed using a Nanodrop spectrophotometer (ND-1000, Nanodrop, Labtech International, Uckfield, UK) and equalised to a concentration of at least 20 ng/μL. A high-capacity complementary DNA (cDNA) kit (Bio-Sciences, Dublin, Ireland) was used to reverse transcribe RNA samples according to manufacturer’s instructions. Taqman gene expression assay for rat cnr1 (assay ID Rn00562880_m1), cnr2 (assay ID Rn03993699_s1), faah (assay ID Rn00577086_m1), mgll (assay ID Rn00593297_m1), and ppara (assay ID Rn00566193_m1) containing forward and reverse primers and an FAM-labelled MGB Taqman probe (Applied Biosystems, Warrington, UK) were used to quantify CB_1_, CB_2_, FAAH, MAGL, and PPARα, respectively, on a StepOne Plus Real-Time PCR System (Applied Biosystems, Warrington, UK). VIC-labelled beta-actin (assay ID Rn00667869; Applied Biosystems) was used as the house-keeping gene and endogenous control. The relative expression of target genes to endogenous control was calculated using the formula 2^−ΔCt^, where ΔCt represents the magnitude of the difference between cycle threshold (Ct) values of the target and endogenous control, and the result is expressed as a percentage of the mean value of the saline-treated control group. The results were discarded when the amplification curve did not reach its maximum plateau after 40 cycles. This happened for FAAH expression in fundus and ileum, MAGL expression in ileum, and PPARα expression in fundus, ileum, and colon.

### 4.9. Statistical Analysis

Gastrointestinal motility, body weight, and food intake data were analysed using a one-way repeated measures analysis of variance (ANOVA) with the Greenhouse–Geisser correction. *t*-tests were run for the behavioural, biochemical, and genic data and in each time-point when a significant intra-subject effect of “treatment-time” and/or between-subject effect of “treatment” was observed in the repeated measures ANOVA. Normality was confirmed using Shapiro–Wilk tests. If data were normal, but not homogeneous, an inequality of variances was assumed. When data were found to fail normality in their distribution, and the highest standard deviation was two times higher than the smallest standard deviation of the data set being analysed, a non-parametric Kruskal–Wallis test was used to analyse the data [67]. Significance was set at *p* < 0.05 and all data are expressed as Mean ± SEM. Statistical analyses were performed by the SPSS 28.0 software package (SPSS Inc., Chicago, IL, USA).

## 5. Conclusions

Our results show that, as expected, there are immediate effects on the gastric function after a systemic single administration of 5 mg/kg of cisplatin, manifested as a significant decrease in gastric emptying, which is compatible with the presence of nausea and vomiting in patients and in experimental animals able to vomit (ferret, dog, etc.). One week later, the decrease in MAGL gene expression (probably associated with an increase of 2-AG) in the antrum and the increase in AEA in the ileum suggest that a reduction in gastric and small intestine motility may still be happening, probably contributing to the decreased body weight and food intake observed in these animals. The dose of 5 mg/kg of cisplatin failed to induce mechanical and heat hypersensitivity four days after treatment but, interestingly, we did observe important alterations in the ECS of the PFC and DRGs that resemble those observed in neuropathic pain states. A decrease in MAGL in the PFC that may be affecting the corticoamygdalar circuit and an increase in CB_1_ expression in L4 and L5 DRGs are common features observed in other models of chronic pain (including neuropathic). Indeed, despite the lack of behavioural effects in these animals, cisplatin administration did alter the ECS in key areas of the pain pathway.

In view of our current results, we propose the use of treatments to modulate the ECS to prevent the side effects induced by chemotherapeutic treatment. These cannabinoid-based treatments could be administered just before or after the first (and each) chemotherapeutic cycle to palliate or, better, prevent gastrointestinal and nervous toxicity induced by chemotherapy.

We expect that our data will stimulate additional preclinical studies in order to explore whether pharmacological manipulation of the ECS is able to alleviate or potentially prevent chemotherapy-induced toxicity, as a first step towards more targeted therapies based on ECS modulation in the clinic.

## Figures and Tables

**Figure 1 pharmaceuticals-17-01256-f001:**
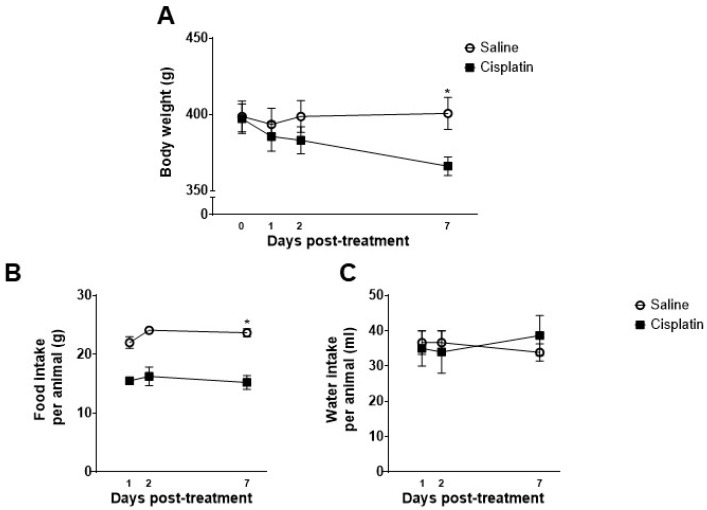
Body weight in grams (**A**), food intake per animal in grams (**B**), and water intake per animal in millilitres (**C**) on day 0, 1, 2, and 7 after treatment with saline (n = 6) or a single dose of cisplatin (i.p., 5 mg/kg; n = 6) in male Wistar rats. Data are expressed as Mean ± SEM of saline- vs. cisplatin-treated animals. Statistical significance set at *p* < 0.05. Repeated measures ANOVA followed by a *t*-test or Kruskal–Wallis in each time-point: * significant difference.

**Figure 2 pharmaceuticals-17-01256-f002:**
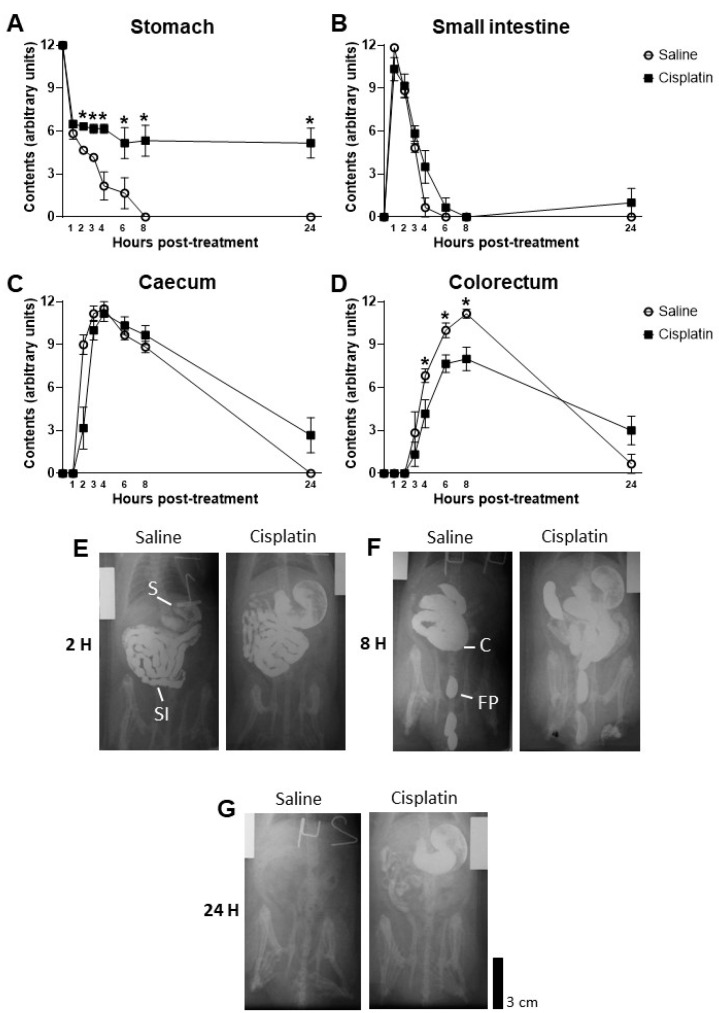
Radiographic analysis of gastrointestinal motility 1, 2, 3, 4, 6, 8, and 24 h after treatment with saline (n = 6) or a single dose of cisplatin (i.p., 5 mg/kg; n = 6) in the stomach (**A**), small intestine (**B**), caecum (**C**), and colorectum (**D**) in male Wistar rats. Contents, in arbitrary units, are expressed as Mean ± SEM of saline- vs. cisplatin-treated animals. Statistical significance set at *p* < 0.05. Repeated measures ANOVA followed by a *t*-test or Kruskal–Wallis in each time-point: * significant difference. Representative X-rays of rats treated either with saline or a single dose of cisplatin (i.p., 5 mg/kg) at time-point 2 h (**E**), 8 h (**F**), and 24 h (**G**); scale bar: 3 cm. S: stomach; SI: small intestine; C: caecum; FP: faecal pellet (in colorectum).

**Figure 3 pharmaceuticals-17-01256-f003:**
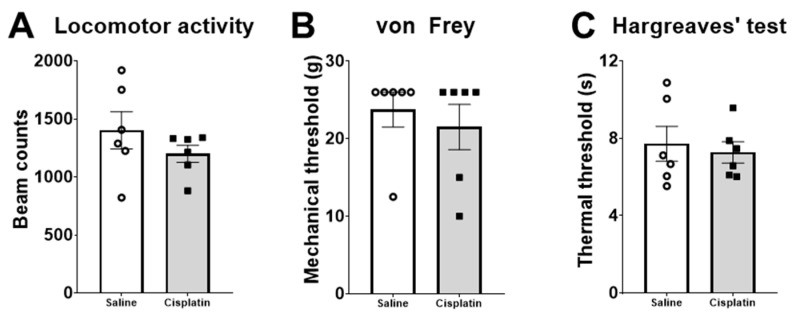
Behavioural analysis 4 days after treatment with saline (n = 6) or a single dose of cisplatin (i.p., 5 mg/kg; n = 6) in male Wistar rats. Locomotor activity in beam counts/30 min (**A**), mechanical tactile sensitivity measured by von Frey filaments as mechanical threshold (grams) (**B**), and heat tactile sensitivity measured by Hargreaves’ test as thermal threshold (seconds) (**C**). Data are expressed as Mean ± SEM of saline- vs. cisplatin-treated animals. Statistical significance set at *p* < 0.05.

**Figure 4 pharmaceuticals-17-01256-f004:**
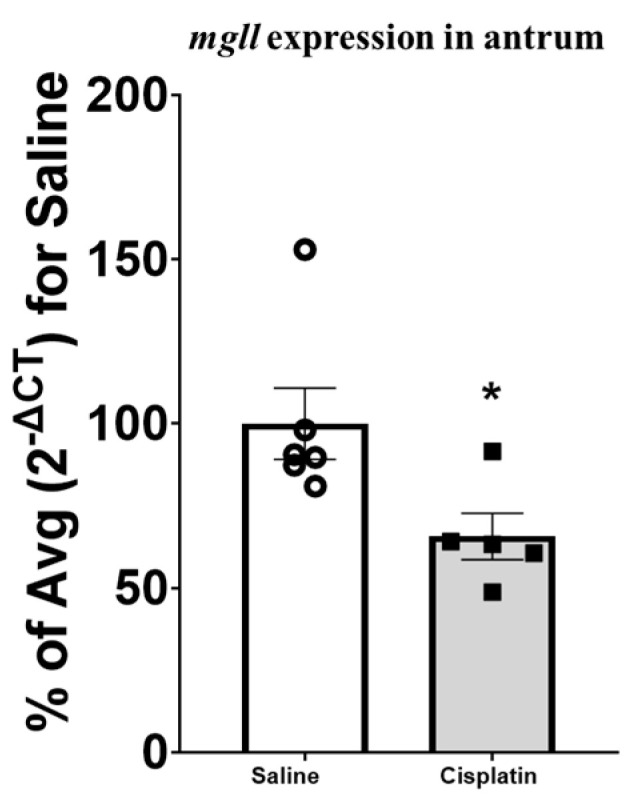
Gene expression of the *mgll* gene in the antrum of male rats after treatment with saline (n = 6) or a single dose of cisplatin (i.p., 5 mg/kg; n = 5). The mean percentage of the saline-treated group (2^−ΔCT^) for saline is expressed as Mean ± SEM of saline- vs. cisplatin-treated animals. Statistical significance set at *p* < 0.05. *t*-test: * significant difference.

**Figure 5 pharmaceuticals-17-01256-f005:**
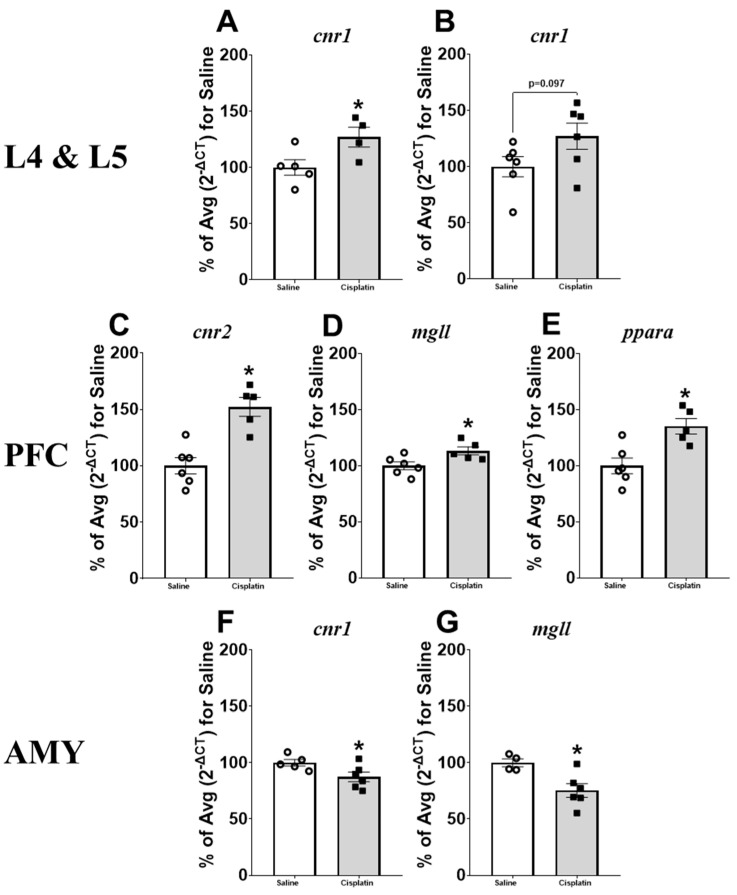
Gene expression in the nervous system of male rats after treatment with saline (n = 5–6) or a single dose of cisplatin (i.p., 5 mg/kg; n = 4–6). Levels of *cnr1* in the dorsal root ganglia L4 (**A**) and L5 (**B**); levels of *cnr2* (**C**), *mgll* (**D**), and *ppara* (**E**) in the prefrontal cortex (PFC); and levels of *cnr1* (**F**) and *mgll* (**G**) in the amygdala (AMY). The mean percentage of the saline-treated group (2^−ΔCT^) for saline is expressed as Mean ± SEM of saline- vs. cisplatin-treated animals. Statistical significance set at *p* < 0.05. *t*-test: * significant difference.

**Figure 6 pharmaceuticals-17-01256-f006:**
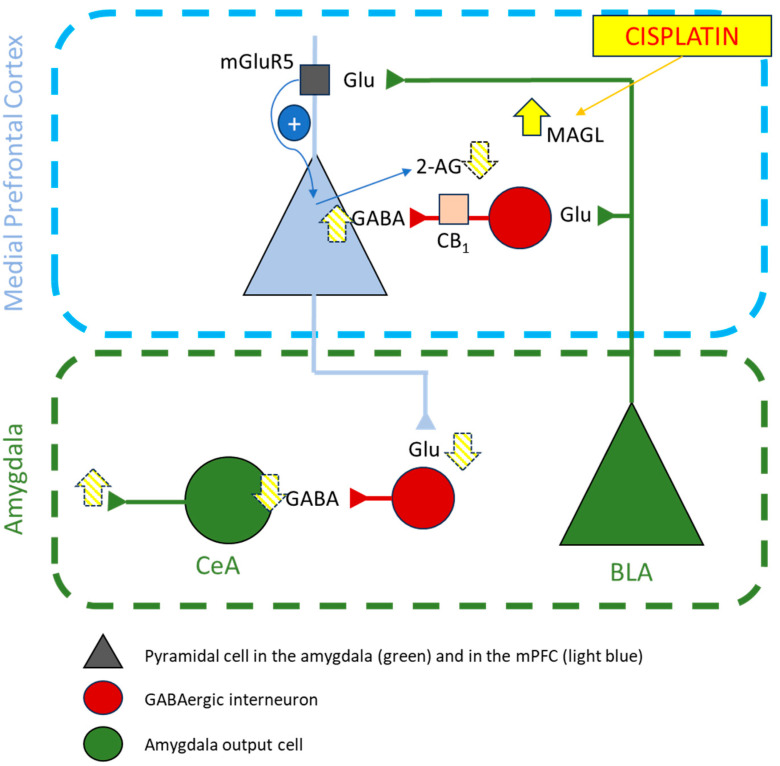
Hypothetical mechanism of the effect of cisplatin on the pain corticoamygdalar pathway. In normal conditions, activation of basolateral amygdala (BLA) pyramidal cells induces the secretion of 2-AG by pyramidal cells in the medial prefrontal cortex (mPFC) thanks to the activation of mGluR5. Administration of 5 mg/kg of cisplatin induced a significant increase in the expression levels of MAGL in the prefrontal cortex, which in turn would deplete the 2AG secreted by prefrontal pyramidal cells and, hence, reduce the activation of CB1 in prefrontal GABAergic interneurons. Prefrontal GABAergic interneurons would then increase their inhibitory influx on mPFC pyramidal cells projecting to GABAergic interneurons in the central nucleus of the amygdala (CeA), which would lead to increased firing of the amygdalar signalling outputs, an effect frequently observed in pain-related emotional responses in animal models of arthritis, colitis, formalin-induced inflammation, and neuropathic pain. The yellow-filled arrow represents the increase in MAGL gene expression observed in the PFC, while yellow-striped arrows represent the hypothetical changes that would follow it in the corticoamygdalar pathway. Schematic representation adapted from [50].

**Figure 7 pharmaceuticals-17-01256-f007:**
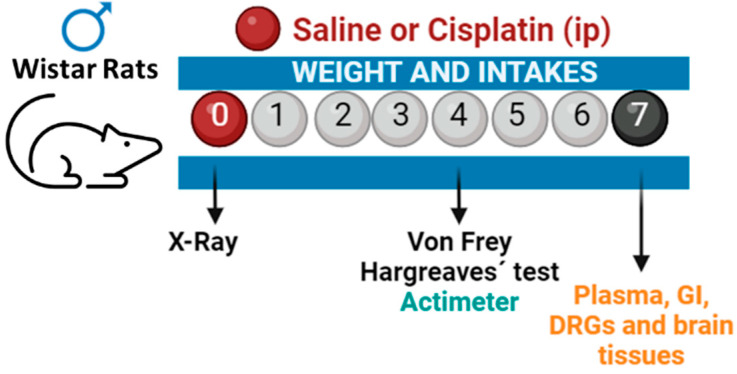
Experimental outline. Male Wistar rats were treated either with cisplatin (i.p., 5 mg/kg; n = 6) or saline (i.p., 0.9% NaCl; n = 6) on day 0. Immediately after, animals were administered barium contrast by gavage and submitted to a radiographic evaluation of gastrointestinal transit during the first 24 h after treatment. Food intake and body weight were recorded on days 1, 2, and 7 and locomotor activity (actimeter) and pain-related behaviours (von Frey and Hargreave’s test) were measured on day 4. On day 7, gastrointestinal and nervous tissues were extracted for further analyses.

**Table 1 pharmaceuticals-17-01256-t001:** Macroscopic analysis of kidneys and gastrointestinal organs of male Wistar rats one week after treatment with saline (n = 6) or a single dose of cisplatin (i.p., 5 mg/kg; n = 6). Data are expressed as mean ± SEM. Statistical significance set at *p* < 0.05. *t*-test: * significant difference.

		Saline	Cisplatin
Weight of animals at sacrifice (g)		400.7 ± 11.6	366.0 ± 6.7 *
Weight of organ at sacrifice (g)	Stomach	6.72 ± 1.17	3.10 ± 0.18 *
	Full small intestine	12.09 ± 0.54	9.51 ± 0.41 *
	Empty small intestine	8.92 ± 0.30	7.51 ± 0.22 *
	Milking	3.01 ± 0.33	1.87 ± 0.26 *
	Caecum	5.45 ± 0.27	5.17 ± 0.46
	Full colorectum	4.36 ± 0.26	4.19 ± 0.40
	Empty colorectum	2.23 ± 0.09	2.04 ± 0.11
	Kidneys	2.54 ± 0.11	2.80 ± 0.20
Area and length of organs at sacrifice	Stomach (cm^2^)	7.05 ± 0.84	4.45 ± 0.28 *
	Caecum (cm^2^)	6.54 ± 0.39	5.83 ± 0.57
	Small intestine (cm)	58.53 ± 1.63	58.01 ± 2.13
	Colorectum (cm)	13.10 ± 0.36	12.06 ± 0.44

**Table 2 pharmaceuticals-17-01256-t002:** Levels of endocannabinoid ligands and related N-acylethanolomines one week after treatment with saline (n = 3–6) or a single dose of cisplatin (i.p., 5 mg/kg; n = 6) in gastrointestinal and central nervous tissues (nmol/g) and plasma (pmol/mL) of male rats. Data are expressed as Mean ± SEM of saline- vs. cisplatin-treated animals. Statistical significance set at *p* < 0.05. *t*-test: * significant difference.

		AEA (nmol/g) Saline vs. Cisplatin	2-AG (nmol/g) Saline vs. Cisplatin	PEA (nmol/g) Saline vs. Cisplatin	OEA (nmol/g) Saline vs. Cisplatin
Gastrointestinal tissue	Antrum	0.025 ± 0.009 vs. 0.016 ± 0.008	74.87 ± 36.73 vs. 32.73 ± 16.41	0.111 ± 0.020 vs. 0.113 ± 0.006	0.736 ± 0.069 vs. 0.621 ± 0.028
	Fundus	0.026 ± 0.012 vs. 0.011 ± 0.002	4.16 ± 1.35 vs. 3.22 ± 0.39	1.872 ± 0.412 vs. 2.044 ± 0.476	3.326 ± 0.482 vs. 3.477 ± 0.273
	Ileum	0.046 ± 0.005 vs. 0.068 ± 0.008 *	90.19 ± 12.01 vs. 82.57 ± 10.70	0.974 ± 0.061 vs. 0.947 ± 0.039	5.435 ± 0.855 vs. 3.987 ± 0.261
	Distal colon	0.028 ± 0.003 vs. 0.023 ± 0.003	23.03 ± 1.87 vs. 19.82 ± 3.00	0.489 ± 0.053 vs. 0.437 ± 0.045	3.31 ± 0.32 vs. 2.69 ± 0.40
Central nervous tissue	Prefrontal cortex	0.014 ± 0.005 vs. 0.017 ± 0.001	12.05 ± 3.12 vs. 11.39 ± 2.03	0.061 ± 0.017 vs. 0.060 ± 0.008	0.087 ± 0.018 vs. 0.088 ± 0.006
	Periaqueductal grey	0.016 ± 0.002 vs. 0.015 ± 0.003	19.17 ± 3.84 vs. 26.97 ± 5.47	0.736 ± 0.105 vs. 0.626 ± 0.113	0.485 ± 0.051 vs. 0.444 ± 0.074
	Amygdala	0.027 ± 0.002 vs. 0.022 ± 0.002	29.05 ± 2.27 vs. 26.40 ± 4.07	0.091 ± 0.007 vs. 0.097 ± 0.008	0.134 ± 0.006 vs. 0.133 ± 0.008
		AEA (pmol/mL) Saline vs. Cisplatin	2-AG (pmol/mL) Saline vs. Cisplatin	PEA (pmol/mL) Saline vs. Cisplatin	OEA (pmol/mL) Saline vs. Cisplatin
Plasma		0.760 ± 0.245 vs. 1.113 ± 0.158	0.878 ± 0.250 vs. 2.269 ± 0.526	9.18 ± 2.32 vs. 12.33 ± 0.66	15.22 ± 4.11 vs. 23.34 ± 1.51

Abbreviations: AEA (anandamide), 2-AG (2-arachidonoylglycerol), PEA (palmitoylethanolamine) and OEA (oleoylethanolamine).

## Data Availability

The raw data supporting the conclusions of this article will be made available by the authors, without undue reservation.

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
