# Peer review of "The Endocannabinoid System of the Nervous and Gastrointestinal Systems Changes after a Subnoxious Cisplatin Dose in Male Rats"

_pharmaceuticals, 2024, doi:10.3390/ph17101256_

Round 1

Reviewer 1 Report

Comments and Suggestions for Authors

In general the article is interesting, it presents a study of the side effects of a subonocova dose (5mg/kg) of cisplatin in a murine model, specifically with the changes in the endocannabinoid system of the nervous and gastrointestinal systems. Some methodological aspects need to be clarified in order to be able to see the veracity of the statistical analyses performed and the discussion generated by this:

What was the age of the rats used? Was it something that was controlled in the study? Because the differences with other studies may be due to the age of the animals. 

How many animals were used in each group (with treatment or with saline)? Because it is not mentioned and it is important to see the statistical analyses performed, and if they were enough to be able to generate this type of conclusions, especially with an in vivo model.  And therefore in the legends of the figures and tables it is necessary to put the n used. 

The conclusion should be expanded more with the results found and what the authors suggest.

In supplementary material, the protocol approved by the Animal 749 Ethics Committee at URJC and Comunidad Autónoma de Madrid (PROEX 142.8/21) could also be included.

Reviewer 2 Report

Comments and Suggestions for Authors

This is a very carefully conducted study aimed at exploring physiological changes occurring in male rats treated by a single subnoxious dose of cisplatin, with special interest in modifications of endocannabinoid system (ECS). The group of authors displays a solid and long experience both in the field of side effects generated by chemotherapeutic agents and in the biology of ECS, as evidenced by a number of references appropriately cited in the text.

Among observed changes, cisplatin induced modifications of body weight and food intake (which were apparently and obviously related), as well as alterations of gastrointestinal motility. Warning: lines 116 to 121, Figures 2A,2B,2C should be Figures 1A, 1B,1C.

On the other hand, the single relatively low dose of cisplatin used in the present study did not evoke behavioral modifications nor neuropathic pain, in agreement with the majority of previously published studies.

The most delicate part of the manuscript concerns exploration of ECS using two appropriate and apparently well-conducted approaches: qRT-PCR for gene expression, LC-MS for endocannabinoid quantification. My remarks will concern 3 points in that part of the manuscript.

1. In the gastrointestinal tract, no significant changes were observed for three receptors (CB1, CB2 and PPARa), whereas MAGL expression was significantly reduced (by one-third) in antrum (Figure 4 and Table S1). However, this was not accompanied by the expected increase in 2-AG content (there was instead a non-significant decrease of 2-AG, see Table 2). The authors provide an explanation for this discrepancy in lines 291-295. I suggest another one, dealing with the fact that 2-AG determination in tissues must always be taken with caution owing to rapid metabolic modifications occurring postmortem, as shown in the brain (see for instance Sugiura et al, Neurosci Lett 2001;297(3):175-8, doi: 10.1016/s0304-3940(00)01691-8 and Brose et al, Lipids 2016;51(4):487-95. doi: 10.1007/s11745-016-41).

2. On the other hand, anandamide (AEA) was significantly increased in ileum (Table 2) but in this case FAAH expression was not reported (Table S1). This avoids a discrepancy similar to that mentioned above, but the authors should at least explain why these data are not available.

3.The remark made in Point 1 above concerning determination of 2-AG and acylethanolamines could also be made for data in Table 2 dealing with central nervous system, where significant differences were not detected. In other words, all the rationale of the effects of cisplatin  on the pain corticoamygdalar pathway is only based on data concerning RNA levels of enzymes and receptors of the ECS. This rationale is partly supported by these data and is well illustrated in Figure 6, which is elegantly adapted from a similar Figure in reference 50. I just advise to be more careful in the conclusions and make two suggestions: 1) title of Figure 6 could be changed into "Hypothetical mechanism of the effect of cisplatin on the pain corticoamygdalar pathway"; 2) in the last paragraph of the Discussion (lines 497-502), I would advice to be more cautious: "can" (line 498) should be changed into "could"; last sentence, I suggest something like: "our data should stimulate additional preclinical studies in order to explore whether pharmacological manipulation of ECS is able to alleviate or potentially prevent chemotherapy-induced toxicity".
